# Dynamic Multilevel Regulation of EGFR, KRAS, and MYC Oncogenes: Driving Cancer Cell Proliferation Through (Epi)Genetic and Post-Transcriptional/Translational Pathways

**DOI:** 10.3390/cancers17020248

**Published:** 2025-01-14

**Authors:** Mario Seres, Katarina Spacayova, Zdena Sulova, Jana Spaldova, Albert Breier, Lucia Pavlikova

**Affiliations:** 1Institute of Molecular Physiology and Genetics, Centre of Bioscience, Slovak Academy of Sciences, Dúbravská Cesta 9, 84005 Bratislava, Slovakia; mario.seres@savba.sk (M.S.); spacayova1@uniba.sk (K.S.); zdena.sulova@savba.sk (Z.S.); 2Department of Molecular Biology, Faculty of Natural Sciences, Comenius University, Ilkovičova 6, 84215 Bratislava, Slovakia; 3Institute of Biochemistry and Microbiology, Faculty of Chemical and Food Technology, Slovak University of Technology, Radlinského 9, 81237 Bratislava, Slovakia; jana.spaldova@stuba.sk

**Keywords:** MYC, EGFR, KRAS, epigenetic and non-epigenetic regulation

## Abstract

This review presents an overview of recent findings on various epigenetic and post-transcriptional modifications of the epidermal growth factor receptor (EGFR), which activates the oncogenes *RAS* and *MYC*. Alterations in EGFR, RAS, and MYC play a crucial role in dysregulated processes that contribute to oncogenesis. Focused, targeted research aimed at understanding the distinct effects of these oncogenes and their carcinogenic modifications has facilitated the successful treatment of various malignancies using inhibitors of EGFR tyrosine kinase activity and other therapeutic agents.

## 1. Introduction

Neoplastic transformation is a multistep process, in which normal regulatory controls on cell proliferation and cell–cell interactions are lost, transforming normal cells into cancerous ones. The tumorigenic process involves the interplay between oncogenes and tumor suppressor genes. Multiple oncogenes, along with mutated apoptotic or tumor suppressor genes, act either simultaneously or synergistically to promote cancer development. When mutations occur in normal genes that promote cell growth and lead to their upregulation (gain-of-function mutations), these genes predispose the cell to cancer and are referred to as oncogenes [1]. During tumor progression, these gene mutations are often involved, with the mutated genes becoming highly overexpressed [2]. Normally, most cells undergoing such changes are programmed to undergo rapid cell death (apoptosis) to prevent the spread of damaged cells. However, if this key apoptotic function is altered and does not function properly, some transformed cells may evade the apoptotic cell death program, leading to cancer progression, as apoptosis is a conditio sine qua non for halting neoplastic disease [3]. In such cases, newly activated oncogenes can cause cells that would normally undergo apoptosis to survive and proliferate instead [4]. Most oncogenes arise from normal genes involved in cell growth, proliferation, or the inhibition of apoptosis, known as proto-oncogenes [1,2]. In tumors, oncogenes may be constitutively expressed, leading to a physiologically and functionally active set of genes whose protein products can exert pleiotropic effects on various regulatory and metabolic pathways within the cell [5].

Cell growth in many types of epithelial cell-derived tumors depends on the activation of signaling pathways controlled by the epidermal growth factor receptor (EGFR) [6]. In this review, we focus on the RAS (rat sarcoma virus) family of proto-oncogenes, MYC (myelocytoma proto-oncogene), and EGF (epidermal growth factor), which are key components of this critical pathway [7]. The proper activity of these elements within the cell is essential, and thus changes in their regulation at both epigenetic and non-epigenetic levels can significantly impact the effectiveness of treatment.

Epigenetic reprogramming involves metabolites that may be closely linked and may regulate each other, thereby contributing to tumorigenesis, invasive trafficking, and immune evasion. For example, the metabolites produced may participate in the regulation of epigenetic mechanisms as substrates or cofactors, and changes in epigenetic modifications in turn regulate the expression of metabolites [8]. However, these changes in the metabolon could induce large-scale epigenetically distinct pathways, encompassing genomic alterations as well as (post)transcriptional and (post)translational processes.

The aim of this review is to summarize and logically connect current knowledge regarding the regulation of these oncogenes.

The EGFR (the epidermal growth factor receptor) signaling pathway operates through both cytoplasmic and nuclear mechanisms [9]. The cytoplasmic pathway is mediated by the phosphorylation of four key modules: RAS (a small GTPase family first identified in this context), PI3K (phosphoinositide 3-kinase), PLC-γ (phospholipase C gamma), and STAT3 (signal transducer and activator of transcription 3). These components regulate gene expression and influence chemoresistance, radioresistance, metastatic potential, and overall cell proliferation. The nuclear pathway is initiated by the binding of EGF as a ligand, or in combination with vitamin D, leading to the nuclear translocation of EGFR. Within the nucleus, EGFR interacts with transcription factors such as E2F1 (E2F transcription factor 1) and STAT3. This interaction results in increased expression of cyclin D1 and B-Myb (Myb-related protein B), promoting accelerated cell cycle progression from the G1 to S phase, along with upregulation of iNOS (inducible nitric oxide synthase), which is associated with enhanced tumor cell proliferation and metastatic capacity (Figure 1) [10].

Due to these complex regulatory effects, EGFR interferes with multiple regulatory pathways, and its aberrant function may be involved in carcinogenesis, acceleration and dysregulation of cell proliferation, epithelial–mesenchymal transition, metastasis, and more [11,12].

## 2. Regulation of the EGFR Oncogene

The epidermal growth factor receptor (*EGFR* gene product) belongs to the receptor tyrosine kinase (RTK) family, which includes four members: EGFR (ERBB1, also known as HER1, or human epidermal growth factor receptor), ERBB2 (HER2, non-ligand-binding), ERBB3 (HER3), and ERBB4 (HER4) [13]. These receptors are single-chain transmembrane glycoproteins consisting of an extracellular ligand-binding domain, a transmembrane domain, a short juxtamembrane domain, a tyrosine kinase domain, and a tyrosine-rich C-terminus. EGFR functions as a receptor for various epidermal growth factors or extracellular protein ligands that interact with its extracellular domain (Figure 2). Upon ligand binding, the receptor undergoes homodimerization and autophosphorylation at multiple tyrosine residues, initiating its signaling cascade [14]. As a proto-oncogene, *EGFR* plays a crucial role in its signaling pathways. The phosphorylated receptor activates mitogen-activated protein kinase (MAPK), protein kinase B (AKT), and extracellular signal-regulated kinase (ERK) pathways, leading to cell proliferation, invasiveness, metastasis (pro-survival effects), or apoptosis [15].

### 2.1. EGFR Gene Mutation

The organization of the *EGFR* gene, located in the p14–p12 region of human chromosome 7 [16], is shown in Figure 2. Different parts of this gene are sites for mutations that can cause *EGFR* to switch from a proto-oncogene to an oncogene. Further mutations in this gene may occur as a result of intensive therapy. It is important to distinguish between genetic mutations (substitutions, insertions (ins), or deletions (del)) that exist prior to treatment and those that occur in response to treatment with tyrosine kinase inhibitors (TKIs) or humanized monoclonal antibodies against EGFR. While the former may not affect the sensitivity of cells to treatments targeting EGFR and its tyrosine kinase activity, the latter have the potential to reduce the efficacy of treatment with TKIs and, sometimes, with humanized monoclonal antibodies against EGFR [17].

The first type includes the so-called activating mutations of *EGFR*, such as deletion of exon 19 (Ex19del) [18] or substitution of L858R in exon 21 [19] (the positions of these loci are shown in Figure 2), which are frequently found in non-small-cell lung cancer (NSCLC) tissue. Exons 18–21 encode the intracellular tyrosine kinase (TK) domain [20]. The therapeutic use of tyrosine kinase inhibitors (TKIs) has been investigated in the case of Ex19del, where more than 50 different deletions have been identified in this coding region, with the most common deletion being E746_A750. Ex19del mutations occur on the β3 strand of the protein, which is part of the tyrosine kinase domain. It is hypothesized that shortening the length of this strand due to the deletion may increase tyrosine kinase activity. Suppression of TK activity can be achieved with first-generation TKIs, such as gefitinib and erlotinib, or second-generation inhibitors, such as afatinib [21].

The second most common *EGFR* mutation is the L858R missense mutation in coding region 21 [22,23]. This mutation substitutes a lipophilic, uncharged leucine residue with a positively charged arginine. Amino acid position 858 is located in the activation loop, where it normally interacts hydrophobically with residues at the N-terminus only when the receptor is in its inactive conformation. Due to the different properties of arginine, this mutation disrupts the activity-suppressing effect, resulting in the kinase domain being locked in a constitutively active state. The mutant protein exhibits up to a 20-fold increase in TK activity compared to the wild type, leading researchers to conclude that *EGFR* L858R expression in lung cancer cells promotes cell invasiveness [24].

### 2.2. Epigenetic Regulation of EGFR

Mutations in the *EGFR* gene linked to resistance to targeted TKI therapy are often caused by disruptions in interconnected epigenetic changes rather than specific genetic mutations. In a study of 47 samples from patients with head and neck squamous cell carcinoma, MS-PCR detected *EGFR* promoter methylation changes [25]. The analysis revealed a notable decrease in methylation in 21% of samples, while 79% retained promoter methylation. In comparison, the control group showed methylation in 90% of samples, confirming a significant reduction in methylation levels in patients with retained methylation. Mutlu et al. reported that this differential methylation profile may vary with age, gender, tissue type, and tumor characteristics [25].

*EGFR* is overexpressed in gastric cancer. In another study, the EpiTYPER assay was used to examine promoter methylation in two *EGFR* regions [26]. The first region included 7 CpG sites (cytosine per guanine methylation site), and the second had 17 CpG sites. Both regions showed hypermethylation in over 90% of cancer cell samples. Quantitative RT-PCR revealed abnormal *EGFR* expression in 33% of samples. Thees data confirmed a significant relationship between *EGFR* expression and promoter methylation, indicating that hypermethylation stabilizes and upregulates EGFR in gastric cancer. High EGFR levels are linked to an elevated risk of invasion and metastasis [26], while EGFR inhibition reduces cancer cell proliferation, migration and angiogenesis, and enhances apoptosis in solid tumors [27].The findings of Weng et al. [26] that hypermethylation of the *EFGR* promoter is associated with its increased expression is unusual. Typically, hypermethylation associated with transcriptional silencing and in neoplastic tissue is more common for tumor suppressor genes. The authors themselves point out this paradox and speculate on possible mechanisms by which promoter hypermethylation could lead to increased gene expression. One of their suggestions is that the promoter methylation near the transcription start site induces changes in the 3D structure of chromatin, thereby increasing transcription. Another possibility is that methylation may inhibit repressors that normally suppress gene transcription [26].

### 2.3. EGFR Post-Translational Modifications

In addition to autophosphorylation, EGFR also undergoes several other post-translational modifications (PTMs) such as S-palmitoylation, S-nitrosylation, methylation, etc. (reviewed in [28]). An example is the methylation modification mentioned below. The protein lysine methyltransferase known as Wolf-Hirschhorn Syndrome Candidate 1-Like 1 (WHSC1L1) typically dimethylates lysine 36 on histone 3 (H3K36), facilitating the conversion of heterochromatin into transcriptionally active euchromatin [29]. In addition, WHSC1L1 may play a role in several other processes, including alternative splicing, transcriptional repression, DNA repair and recombination. In a study of post-translational modification of EGFR [30], WHSC1L1 was found to monomethylate lysine 721 in the TK domain. This methylation enhances the activation of downstream ERK cascades, leading to increased DNA synthesis and cell cycle progression in head and neck squamous cell carcinoma cells [30].

## 3. Modification of the RAS Oncogene Family

The RAS gene superfamily comprises over 170 RAS related proteins [31]. In humans, there are 35 members. RAS proteins are small GTP-binding proteins that play essential roles in cell signaling and communication [32]. While RAS proteins share similar functions, their specific roles can vary within cells. They activate adenylate cyclase, leading to cAMP production and initiating various downstream processes. The RAS gene family includes major oncogenes such as Kirsten RAS viral oncogene homolog (KRAS), Neuroblastoma RAS (NRAS), and Harvey RAS viral oncogene homolog (HRAS) [31,32]. These genes are crucial for key cellular signaling pathways that regulate the cell cycle, cytoskeletal dynamics, differentiation, proliferation, survival, molecular and vesicular transport, and cell growth [33] (Figure 3).

In order for cancer cells to survive and grow, they must evade the immune system. RAS reduces the expression of MHC-1 (major histocompatibility complex 1) on the cell surface, making the cells less vulnerable to death induced by cytotoxic T cells [35,36]. RAS also promotes the evasion of cancer cells from the immune system by stabilizing the mRNA of PD-L1(programmed cell death ligand 1) in a MEK (mitogen-activated protein kinase kinase)-dependent manner through downregulation of tristetraprolin (TTP/ZFP36), which binds and degrades mRNA [37].

RAS also supports tumor vascularization and inflammation induced by the secretion of IL-8 (interleukin 8), mediated through the MAPK and IP3K pathways [38]. Tumor vascularization is further supported by KRAS-mediated induction of hypoxic HIF signaling, which increases the expression of VEGF (vascular endothelial growth factor) [39].

### 3.1. RAS Family Genes Mutation and Post-Transcriptional Regulation

RAS family genes are among the most frequently mutated in various diseases, including cancer, making them important prognostic markers for disease progression [32]. In human tumors, *RAS* involvement (*KRAS*, *NRAS*, *HRAS*) is primarily associated with activating mutations at codons 12, 13, and 61 [40]. Of the RAS family members, the *KRAS* oncogene is the most frequently mutated, with mutations occurring in up to 23% of cases [41,42]. Mutated *KRAS* is associated with highly aggressive cancers, including pancreatic, lung, and colorectal cancers. The second most frequently mutated *RAS* oncogene is *NRAS*, found mutated in many cancers but particularly prevalent in melanoma [42,43]. *HRAS*, the least frequently mutated member, is typically associated with genitourinary and head and neck cancers, and has also been detected in acute myeloid leukemia [42].

Common mutations in the *KRAS* gene include amino acid substitutions G12C, G12D, and G12R, each linked to different tumor types. The *KRAS* G12C mutation, involving a glycine-to-cysteine substitution, is particularly significant in non-small-cell lung cancer, where it leads to persistent activation of the KRAS oncogenic pathway, driving uncontrolled cell growth [44]. The most common mutation in *HRAS* is the G12V substitution, while *NRAS* frequently mutates at position 61, with Q61R being a prevalent variant [45].

The small molecule sotorasib has shown inhibitory effects on this pathway by specifically binding covalently and irreversibly to mutant *KRAS* in the switch II region, which is unique to the inactive GDP-bound conformation of *KRAS* G12C [46]. This binding stabilizes *KRAS* G12C in its inactive form, preventing oncogenic KRAS signaling. In a study with 126 patients carrying *KRAS* G12C mutations, 37.1% responded positively to sotorasib treatment in combination with docetaxel [46]. Responses to sotorasib treatment lasted an average of 11 months, with tumor shrinkage observed in most cases [46].

Another KRAS inhibitor, such as adagrasib (MRTX-849) or adagrasib combined with TNO155 (SHP2i), is used in the treatment of the *KRAS* G12C mutation. The development of KRAS inhibitors has led to significant therapeutic progress, but resistance continues to limit their efficacy. Synthetic lethal (SL) genes implicated in resistance to *KRAS* G12C inhibitors (G12Ci) in non-small-cell lung cancer (NSCLC) include serine-threonine kinases, tRNA- and proteoglycan-modifying enzymes, and components of the YAP (Yes-associated protein 1)/TAZ (YAP homologue)/TEAD (transcriptional enhanced associate domain) pathway. The YAP/TAZ/TEAD pathway has been extensively validated in vitro and in mouse models using siRNA/shRNA (small interfering/short hairpin RNA) experiments. For instance, YAP, a key effector protein of the Hippo pathway, is a transcriptional coactivator that regulates chromatin activation and the expression of cell cycle genes, promoting cell growth and migration [47,48].

RNA interference by microRNAs (miRNAs) is another method to regulate *RAS* expression and the downstream processes it controls, such as promoting cell growth or, conversely, inducing apoptosis [49]. One of the first and most significant miRNAs identified in this context is let-7, which plays a key role in regulating RAS family genes [50]. While let-7 family members are upregulated during cell differentiation, numerous studies have reported a downregulation of let-7 in various tumor types [51,52]. This miRNA has also demonstrated prognostic significance in non-small-cell lung cancer [53].

Major members of the RAS superfamily, such as KRAS, HRAS, and NRAS, are regulated by miR-181 [54]. The downregulation of miR-181 is a known mechanism that leads to the oncogenic activation of RAS. MiR-181 has been observed to be downregulated in several cancers, including oral squamous cell carcinoma [55,56], gastric cancer [57], and glioma [58], suggesting that miR-181 downregulation contributes to oncogenic RAS activation in these tumors.

Other miRNAs involved in the regulation of RAS family members include miR-18a-3p and miR-143 in colorectal cancer, as well as miR-217, which directly interacts with *KRAS* mRNA by binding to the 3’-untranslated region, leading to its degradation or inhibition of translation [59,60,61].

### 3.2. Epigenetic Regulation Members of RAS Family

As increasing evidence links tumorigenesis to cellular changes driven by epigenetic modifications, it is now essential to consider epimutations alongside genetic mutations in understanding oncogenic mechanisms [62]. Epigenetic alterations in the *HRAS* gene have been observed in urinary tract and bladder tumor tissue, where CpG islands in the 5’ promoter regions are typically unmethylated, allowing potential access to transcription factors [63]. Numerous studies have confirmed that hypomethylation in these promoter regions, coupled with overexpression of non-mutated RAS proteins, contributes to the transformation of healthy cells into tumor cells. Aberrant methylation can result from mutations in genes encoding DNA methyltransferases (DNMTs) or from external factors such as diet, tobacco, or arsenic exposure, which may influence the methylation profile. Hypomethylation of CpG islands in *RAS* oncogene promoters, even in the absence of mutations, is characteristic of bladder tumor initiation and progression. Thus, aberrant methylation of *RAS* genes serves as an early biomarker of bladder tumorigenesis [64].

Methylation of *RAS* gene promoters is a key factor in RAS protein expression. Biopsies from bladder cancer patients have shown a significant positive association between hypomethylation of *KRAS*, *NRAS*, and *HRAS* promoters and their expression levels [65]. The nucleotide sequence at the 5’-end of the distal *KRAS* promoter contains numerous CpG islands, lacks TATA and CCAAT boxes, and has sequence similarities to other housekeeping genes, such as dihydrofolate reductase (DHFR). Additionally, it contains binding regions for anchorage-specific protein 1 (Sp1) and neurofibromin 1 (NF1), which are associated with transcription initiation complexes [66,67].

Studies on the hypomethylation of *RAS* oncogene promoters have confirmed that overexpressed, non-mutated RAS protein can contribute to tumorigenesis and that altered methylation patterns influence its expression. Methylation analyses of the *HRAS*, *KRAS*, and *NRAS* promoters were conducted using methylation-specific PCR (MSP), which employs specific primers to distinguish between methylated and unmethylated DNA [65]. The CpG islands in these gene promoters were significantly, though not uniformly, hypomethylated. In cancer patient samples, all three promoters showed reduced methylation compared to healthy tissue controls. The results indicated that extensive CpG hypomethylation in the *NRAS* and *KRAS* loci is essential for their overexpression in tumor cells [65]. The same authors have shown that increased hypomethylation in the *HRAS* promoter was inversely associated with lower expression in bladder cancer, suggesting that other genetic or epigenetic factors may contribute to HRAS-mediated tumorigenesis.

Hypomethylation of *RAS* oncogene promoters has also been observed in hepatitis C virus (HCV)-induced hepatocellular carcinoma (HCC) [68]. In this study, MSP confirmed that the *NRAS* promoter was significantly hypomethylated compared to that in healthy tissue. Further, NRAS protein expression, measured via qRT-PCR in the Huh-7 cell line, showed increased expression after transfection with HCV core, NS5a, and NS2 viral genes. This finding suggests that HCV genes may heighten malignancy risk, making hypomethylated *NRAS* a potential epigenetic biomarker for HCC progression in hepatitis C patients [68].

A deeper understanding of DNA methylation events linked to oncogenic *KRAS* expression is provided by a study examining baseline DNA methylation at CpG islands in 11 KRAS-dependent pancreatic cancer cell lines with *KRAS* mutations, which revealed strikingly similar methylation patterns [69]. *KRAS* knockdown resulted in unique methylation changes, showing limited overlap between cell lines and affecting over 8000 differentially methylated (DM) CpGs. In contrast, treatment of *KRAS*-mutant Pa16C pancreatic cancer cells with the selective ERK1/2 inhibitor SCH772984 produced fewer than 40 DM CpGs. This finding suggests that ERK, a primary downstream effector of *KRAS*, is not a broadly active driver of *KRAS*-associated DNA methylation and that these methylation events are largely independent of ERK signaling [69].

### 3.3. RAS Family Post-Translational Modifications

Proteins translated from RAS templates undergo numerous PTMs that modify their activity or are essential for membrane targeting [70]. Since PTMs are catalyzed by enzymes, these enzymes represent critical targets for drug discovery—an area of high priority in RAS-focused research (Figure 4). Reversible phosphorylation of signaling proteins is a particularly common mechanism for regulating signaling pathways. Although the phosphorylation–dephosphorylation cycle has been extensively studied, its full functional implications remain incompletely understood. The involvement of PKC in KRAS4B phosphorylation was first reported 33 years ago [71], and the effects of serine 181 phosphorylation of this protein by both PKC and cyclic GMP-dependent protein kinase 2 continue to be actively investigated (reviewed in [70]).

Similarly, HRAS has been identified as a substrate for protein kinase A (PKA), although the physiological significance of this modification remains unclear [72]. However, phosphorylation of HRAS by glycogen synthase kinase 3β (GSK3β) at threonine residues 144 and 148 results in polyubiquitination, mediated by the F-box protein β-transducin repeat-containing protein (β-TrCP, which serves as a substrate adaptor for the SCF E3 ubiquitin ligase complex [73]), leading to subsequent degradation [74].

More recently, Yin et al. reported that NRAS is phosphorylated at S89 by serine/threonine kinase 19 (STK19), a modification that activates NRAS signaling [75]. Inhibitors of STK19 have shown efficacy in a mouse model of NRAS-induced melanoma.

Ubiquitination can regulate multiple RAS interactions. The primary sites of monoubiquitination are located at position 147 in KRAS [76,77] and position 117 [76] in HRAS. Monoubiquitination of KRAS at site 147 increases RAS activity by impairing GTPase-activating protein (GAP) function [78,79]. GAP normally accelerates the hydrolysis of GTP to GDP, effectively “turning off” RAS signaling by converting it from its active, GTP-bound state to its inactive, GDP-bound state [80]. This GAP defect results in increased MAPK activation, signaling, and tumorigenesis [78,79]. In contrast, monoubiquitination of HRAS at position 117 promotes rapid GDP-to-GTP exchange and activates RAS by an independent manner, on a guanine nucleotide exchange factor (GEF) [81]. GEFs are proteins that activate RAS by facilitating GDP-GTP exchange, converting RAS to its active, GTP-bound state [82]. Additionally, Rabex-5, a guanine nucleotide exchange protein, promotes mono- and diubiquitination of HRAS and NRAS, leading to their relocalization to endosomes and reduced MAPK signaling [81]. Rabex-5 is specific for RAB5, a small GTPase involved in early endocytosis. RAS degradation is regulated by ubiquitination through proteins such as leucine zipper-like transcription regulator 1 (LZTR1), β-TrCP1, and SMAD ubiquitination regulatory factor 2 (SMURF2), which direct RAS to the proteasome and autolysosomes, thereby reducing MAPK signaling [81]. In contrast, the deubiquitinase OTUB1, commonly overexpressed in non-small-cell lung cancer, removes ubiquitin from RAS, promoting plasma membrane localization and enhancing MAPK signaling [81,83].

Another PTM involves the covalent attachment of small ubiquitin-like modifier (SUMO) peptides to client proteins, a process known as sumoylation. In humans, there are five SUMO isoforms. SUMO1–SUMO3 are ubiquitously expressed and expression of SUMO4 and SUMO5 is restricted to specific tissues [84]. They are small, ubiquitin-like proteins with a length of about 100 amino acids and a mass of about 12 kDa [85]. Unlike ubiquitination, sumoylation does not mark proteins for degradation. Sumoylation occurs at lysine residues on the client protein, specifically at a recognition site with the sequence ψKxE (where ψ is a hydrophobic residue, K is lysine, x is any amino acid, and E is glutamate) [86]. For RAS proteins, sumoylation takes place at the conserved lysine residue in position Lys42 [87], located near the effector region (residues 30–40). Sumoylation serves as a key mechanism in regulating RAS functions in cell proliferation, differentiation, and malignant transformation, highlighting an area for further focused research (reviewed in [88]). The three main RAS isoforms (HRAS, KRAS, and NRAS)undergo sumoylation with different SUMOs, most commonly with SUMO3. Modification of RAS by SUMO3 appears to promote activation, as inhibiting sumoylation or mutating Lys42 leads to reduced RAS signaling and decreased migration and invasiveness across multiple cell lines [81]. Substituting lysine at position 42 with arginine in the *KRAS* gene inhibited cell migration and invasion in vitro across multiple cell lines, including transformed pancreatic cells [89]. This modification impaired activation of the RAF/MEK/ERK signaling pathway and led to reduced phosphorylation of various other protein kinases, such as c-Jun N-terminal kinase, checkpoint kinase 2, and focal adhesion kinase.

Interestingly, the small GTP-binding proteins RAS, RAP, and RAC are targets of the lethal toxin from *Clostridium sordellii*, which induces specific glucosylation [90]. K/H/NRAS isoforms are glucosylated at threonine 35 [91], located within the RAS effector domain. This glucosylation has minimal impact on GTP binding but significantly reduces GTPase activity, as it prevents the interaction of p120GAP and neurofibromin (NF-1) with glucosylated RAS. The net effect of toxin-induced glucosylation is a complete blockade of downstream RAS signaling [92].

To ensure the proper function of KRAS, HRAS, and NRAS, a prenylation PTM is required to regulate their trafficking and target localization [93]. Prenylation of RAS proteins occurs in the cytosol, allowing them to attach to the cytosolic side of the endoplasmic reticulum (ER) membrane. This modification takes place at the cysteine SH group of the CAAX motif—where C is cysteine, A represents aliphatic amino acids (like leucine, valine, or isoleucine), and X can be any amino acid [94]. This motif is located at the C-terminus of RAS proteins. In the ER, further processing involves cleavage of the AAX sequence from the prenylated cysteine by RAS-converting enzyme 1 (RCE1) [95], after which isoprenylcysteine is methylated by carboxyl methyltransferase (ICMT) at the liberated COOH group of cysteine [96]. Processed NRAS and HRAS then migrate to the Golgi apparatus, where they attach to the cytosolic side of the Golgi membrane. Here, they undergo palmitoylation, which increases their affinity for phospholipid membranes, and are transported by vesicular transfer to the plasma membrane [93]. NRAS can also be shuttled through the cytosol by chaperones such as PDE6δ and VPS35 [97]. At the plasma membrane, depalmitoylation occurs, allowing NRAS and HRAS to cycle back to the Golgi for another round of palmitoylation. KRAS4A is also palmitoylated, though the specific site of this modification has not been determined [93]. At the plasma membrane, KRAS4A is depalmitoylated and relocates to endomembranes, including the outer mitochondrial membrane.

KRAS4B, by contrast, has a strong polybasic region that substitutes for palmitoylation [98]. This prenylated protein interacts with chaperones, such as PDE6δ, to aid its membrane localization, but it loses affinity for the plasma membrane when serine 181 (S181) in its hypervariable region (HVR) is phosphorylated [99]. Whereas a farnesyl-only modification imparts relatively low affinity to membranes, each of these second signals adds the affinity required for trafficking and stable association with membranes [100].

Several GTPases, including RAS, contain a redox-sensitive NKCD motif, which consists of four amino acids: asparagine, lysine, cysteine, and aspartate, located at positions 116–119 (N116-K117-C118-D119) in RAS [101]. Reactive oxygen and nitrogen species can oxidize Cys118, affecting the protein’s stability, activity, localization, and protein–protein interactions. These modifications may also influence the reactivity of other cysteine residues at positions 80, 181, 184, and 186 [101]. GTPases in the RAS superfamily can be regulated by nitric oxide and other reactive oxygen/nitrogen species [102]. When reacting with nitric oxide, RAS undergoes nitrosylation at Cys118. Studies show that RAS nitrosylated at Cys118 retains a similar GEF-binding structure to its non-nitrosylated form [103]. Contrary to the assumption that S-nitrosylation activates RAS, recent evidence indicates that RAS nitrosylation is a reversible modification from NO action, rather than an active state [104].

## 4. Regulation of MYC Oncogene Expression and Activity

The avian myelocytomatosis viral oncogene homolog (*MYC*/MYC) is one of the most frequently mutated oncogenes, and its overexpression is strongly associated with cancer, making it a major focus of research [105]. This critical gene is located on chromosome 8 and regulates up to 15% of all genes [106]. *MYC* has paralogues, such as *L-MYC* and *N-MYC*, located on chromosomes 1 and 2, respectively [107,108]. These proteins share several highly conserved, functionally important regions, which are similarly organized.

*L-MYC* is expressed and active in dendritic cells, gastrointestinal cells, and lung cells. *N-MYC*, on the other hand, is expressed in neural stem cells, where it promotes rapid cell proliferation in stem cells and neuroendocrine tissues in the developing brain [109,110].

As a transcription factor, the MYC protein is involved in the regulation of cell proliferation and apoptosis (Figure 5). The ability of MYC to regulate apoptotic cell death results from the coordinated activation of MYC and several protein partners, such as myc-associated factor X (MAX), which facilitates DNA binding and activates transcription or binding of Max dimerization protein/Max interfactor/Max binding protein antagonists [111]. In hematopoietic stem cells, MYC controls the balance between self-renewal and differentiation [112,113]. However, it is problematic to consider the oncogenic MYCprotein as a therapeutic target because its inactivation may also inactivate physiological MYC in normal tissues, which can negatively affect the physiological state [114]. In addition, MYC is a disordered protein that lacks effective binding pockets on its surface [115] and can therefore adopt multiple conformations, i.e., it exhibits conformational plasticity in the absence of its partners. In the presence of MAX, it forms a heterodimer with MYC and regulates many genes involved in cell cycle progression, neoplastic transformation, proliferation, apoptosis and genomic destabilization, and also blocks differentiation [116]. *MYC* gene regulation can be disrupted by a variety of mechanisms, including gene amplification, chromosomal translocation, or alteration of signaling pathways. Excessive activation of the *MYC* gene by amplification can be observed in breast, pancreatic or lung cancer, while its aberrant influence on signaling pathways such as PI3K/AKT or MAPK is typical of glioblastoma and colon cancer [117].

The most common genetic alteration of the *MYC* gene across various cancers is its amplification, which leads to elevated MYC protein levels, resulting in abnormal activity and, ultimately, cancer development [118]. This oncogenic mutation of the *MYC* gene has been identified in small-cell lung cancer (SCLC) samples. *MYC* gene amplification was detected using chromogenic in situ hybridization (CISH) analysis with a *MYC* probe. In a study of 77 samples, the MYC oncogene was overexpressed in 20% of tumors [119]. The authors confirmed a 100% correlation between *MYC* gene amplification and increased expression. However, elevated *MYC* gene expression was also observed in samples without recorded amplification, suggesting possible epigenetic modifications [119].

### 4.1. Epigenetic Regulation of MYC

In addition to genetic alterations, *MYC* expression is modulated at both the transcriptional and translational levels by a variety of epigenetic modifications. These alterations increase the synthesis of MYC, a critical transcription factor that drives the reprogramming of normal cells into cancer cells, thereby enhancing their proliferation and resistance to chemotherapy [120]. Methylation of CpG sites belonging to the *MYC* gene appears to be higher in DNA from normal tissues than from tumor tissues. For example, in a rat model of liver carcinogenesis induced by prolonged exposure to a choline- and L-amino acid-deficient diet [121], DNA from normal liver hepatocytes (HN) was compared with DNA from hepatocellular carcinoma (HCC) cells. All 33 cytosines in the 5′ upstream region of the –*Myc* gene were completely methylated in HN, whereas these cytosines were completely unmethylated in HCC [122]. This demethylation was associated with a significant increase in *Myc* transcript levels in HCC compared to HN. These findings suggest that hypomethylation of the *MYC* gene may play a role in the malignant transformation of HN to HCC.

Like most oncogenes, the *MYC* oncogene promoter is hypomethylated in tumor cells. To induce hypomethylation of the *MYC* promoter and changes in its corresponding mRNA, Urbanek-Olejnik et al. used the liver carcinogen phenobarbital (PB) and administered it to male rats for 14 days [123]. Methylation of the *Myc* promoter was measured by methylation-specific PCR (MSP), and mRNA and DNMT1 levels were determined by RT-PCR and Western blot. The *Myc* promoter was demethylated in 50% of the samples analyzed after the first dose of PB. Three doses of PB induced even more promoter demethylation. After 14 days of the experiment, the *Myc* promoter was also demethylated in 63% of the samples and no hypomethylation was detected in the control rat samples [123].

In addition to DNA methylation, histone modification is another important epigenetic modification. MYC regulates the expression of target genes in conjunction with its obligatory protein partner MAX, which heterodimerizes and binds to the enhancer box (E-box) on DNA. While the MAX protein lacks a regulatory domain, MYC contains an N-terminal transcriptional activation domain (TAD) that recruits histone acetyltransferases (HATs) such as GCN5 (General Control Protein Non-repressed 5), TIP60 and the p300/CBP multiprotein complex. P300 is a transcriptional co-activator with HAT activity [124,125]. This protein modulates chromatin conformation by acetylating histones, and its overexpression has been observed in several tumor types.

### 4.2. MYC Post-Translational Modifications

Like other proteins, MYC is subject to various PTMs [126,127]. Therefore, several PTM elucidation studies have been carried out to increase our knowledge or to develop drugs for targeted therapy (Figure 6). For example, lysine residues at the N-terminus of the *MYC* oncogene are direct substrates for several acetyltransferases. The MYC HAT p300 acetylation mapping study showed direct acetylation of 6 lysine residues in human and mouse samples, namely K143(K144), K157(K158), K275, K317, K323 and K371 [128]. In in vitro experiments, MYC was acetylated by overexpression of CREB-binding protein (CBP) and GCN5 in cells after culture, affecting its effect on target genes [128].

According to findings by Nebbioso et al. [129], MYC can undergo post-translational methylation, a modification that plays a crucial role in the anti-cancer effects of histone deacetylase (HDAC) inhibitors in acute myeloid leukemia. MYC associates with the promoter region of the tumor necrosis factor-related apoptosis-inducing ligand (TRAIL) at the proximal GC box via SP1 or MIZ1, thereby blocking TRAIL activation. HDAC inhibitors stimulate TRAIL expression by disrupting the binding of MYC to the TRAIL promoter. Specifically, HDACs (such as vorinostat and entinostat) acetylate MYC at lysine 323, which reduces MYC expression at both the mRNA and protein levels [129]. This allows TRAIL to be expressed and exert its pro-apoptotic effect. Importantly, these processes do not occur in normal cells.

Ubiquitination, another type of PTM, also affects the lifespan of the MYC protein. In the normal non-tumor cell, excess MYC protein expression is removed by the ubiquitin–proteasome system, where MYC is rapidly degraded [130]. In tumor cells, the balance between MYC expression and degradation is skewed in favor of expression, leading to accumulation of MYC protein in the cell, its increased stability and hyperactivity. This leads to accelerated cell proliferation, inhibition of apoptosis and, ultimately, cancer initiation and progression [131]. Modified E3 ubiquitin ligases such as FBW7 (F-box and WD repeat domain-containing 7), SKP2 (S-phase kinase-associated protein 2) or TRIM32 (Tripartite motif-containing protein 32), which recognize specific MYC sequences and catalyze the transfer of ubiquitin to MYC, are responsible for the deregulation of these mechanisms [132]. Ubiquitination of MYC can be reversed by deubiquitinating enzymes (DUBs) such as USP7, USP13, USP22, USP28, and USP37 [133]. Recently, USP7 has been shown to stabilize MYC by antagonizing TRIM32-mediated polyubiquitination, and this process is required for neural stem cell maintenance [134]. The DUB enzymes USP22, USP28 and USP37 interact with MYC in the nucleoplasm and USP36 in the nucleolus and reverse Fbw7a-mediated MYC ubiquitination, which is essential for cancer cell proliferation [135,136,137,138]. Overexpression of USP22 leads to slowed growth, migration, and tumorigenesis of breast cancer cells in a MYC -dependent manner [135]. Overexpression of USP28 has been found in colon and breast cancer [139]. High levels of USP36 expression have been found in human breast and lung cancer, suggesting an oncogenic role [140]. USP37 is also upregulated in human lung cancer tissues, and positively correlates with MYC content [136]. FBXL14 ubiquitinates and destabilizes MYC, whereas USP13 is able to stabilize MYC by antagonizing the FBXL14-mediated ubiquitination process.

Recent studies suggest that sumoylation and desumoylation may also be involved in the regulation of MYCprotein activity and stabilization. Interestingly, there is evidence for a correlation between MYCubiquitination and sumoylation. Deregulation of the regulatory network of MYCubiquitination and sumoylation may contribute to tumorigenesis [141]. Mass spectrometry has shown that MYC can be sumoylated on at least 10 lysines (K52, K148, K157, K317, K323, K326, K389, K392, K398 and K430, Figure 5) [142]. However, these authors concluded that mutation of all 10 lysines did not abolish sumoylation of MYC and did not significantly alter its cellular levels or activity. For example, substitution with arginine at position K326R had no apparent effect on MYC stability, intracellular localization and transcriptional targets, or on the biological effects of its overexpression in two different cell systems [143]. From these results, it is hypothesized that sumoylation may represent a quality control feature for misfolded MYC proteins [144]. This suggests that sumoylation of MYC may act in a variety of ways, and leaves open the question of whether sumoylation regulates MYC stability and/or activity. The use of proteasome inhibition significantly induces the level of sumoylation of MYC by either SUMO1 or SUMO2 [145]. The sumo ligase PIAS1 has been shown to sumoylate MYC, mainly at K51 and K52. It was originally thought to play a key role, as its knockdown reduced MYC sumoylation and increased MYC transcriptional activity [142,146]. However, PIAS1-mediated sumoylation increased MYC stability and transactivation activity by recruiting JNK1 (c-Jun N-terminal kinases) to phosphorylate MYC at S62 [145].

MYC is generally recognized as challenging for targeted therapy, due to its nuclear localization and the lack of accessible active sites that would enable binding with a small-molecule ligand. In a study investigating direct therapeutic targeting, researchers monitored the activity of a small peptide, Omomyc, which competes for binding at the E-box on DNA [147]. Omomyc forms a heterodimer with the MAX protein, displacing oncogenic MYC from its interaction with MAX and its subsequent binding to the E-box. This substance has shown therapeutic potential in vivo across multiple models. To examine whether Omomyc also influences epigenetic modifications, researchers transfected 293T cells with vectors expressing both MYC and Omomyc [148]. In this system, Omomyc clearly altered histone 3 acetylation at lysine 9 (H3K9Ac) and methylation in a manner opposite to MYC, thereby acting as its antagonist.

## 5. Translational Research on EFGR, RAS and MYC as Drug Target

EFGR appears to be a suitable target for the treatment of cancer. An overview of clinical trials is provided by the National Cancer Institute (Bethesda, MD, USA) on the website: https://www.cancer.gov/research/participate/clinical-trials/intervention/egfr-targeting-agent (accessed on 7 January 2025). An interesting possibility is to overcome drug resistance by targeting EFGR [149].

As KRAS, NRAS, and HRAS are commonly altered genes in cancer, the pursuit of RAS inhibitors remains an active area of contemporary drug development research. The effectiveness of allele-specific covalent inhibitors targeting the most frequently mutated genes highlights a promising avenue for creating new treatments (reviewed in [150]. Direct suppression of RAS family proteins can be combined with therapies addressing RAS activation pathways.

Like EGFR and RAS, MYC is a viable target for pharmacological modulation, offering promising therapeutic options for cancer treatment. For instance, the Omomyc inhibitor (MYC OMO-103) initially entered Phase 1 [151] clinical trials, which it successfully completed, and it has now advanced to Phase 2 [152].

In the near future, modulators of the EGFR/RAS/MYC axis are likely to become effective therapeutic tools for treating cancer, particularly in cases where existing therapies are unsuccessful.

## 6. Conclusions

The first quarter of the 21st century has brought significant advances in the understanding of the molecular mechanisms that regulate cellular homeostasis and their dysfunction in cancer development. This growing body of knowledge is enabling the development of therapies tailored to the altered phenotypes of a patient’s tumor tissue, offering more precise treatments that target the molecular roots of the disease. Among the regulatory pathways dysregulated in oncogenesis, the EGFR/MYC/RAS axis plays a central role. Its deregulation is often at the core of tumor initiation, progression, and metastatic potential of neoplastic cells in various cancer types. An overview of some of the described molecular modifications of genes and proteins of this axis and their impact on cell proliferation is given in Table 1.

This review highlights key findings on (epi)genetic modifications and post-translational alterations affecting these signal transducers, providing a structured framework for exploring targeted pharmacological interventions in these dysregulated processes. A deeper understanding of how this regulatory axis is perturbed in different cancers underscores the potential of EGFR, MYC and RAS as promising targets for the development of effective treatments for cancer patients.

## Figures and Tables

**Figure 1 cancers-17-00248-f001:**
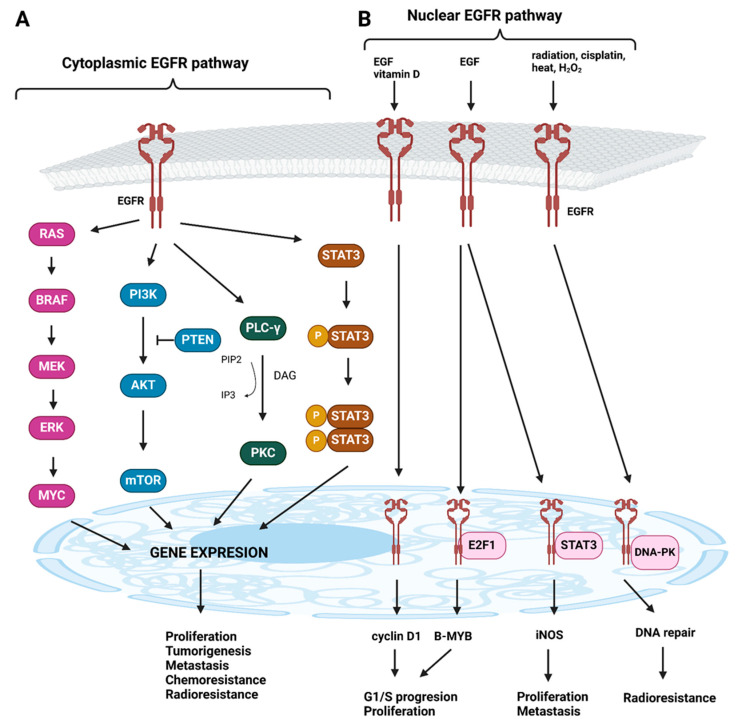
EGFR signaling pathway scheme. (**A**) The cytoplasmic EGFR pathway is consisted of four major modules: RAS-BRAF-MEK-ERK-MYC, PI3K-AKT-mTOR, PLC-γ-CaMK/PKC and STAT3s. Activated pathways ensure the regulation of gene expression of cell proliferation, metastasis, chemoresistance and radioresistance. (**B**) The EGFR nuclear pathway can be activated by various ligands. Nuclear EGFR contributes to the increased expression of cyclin D1 and B-Myb, thereby contributing to the acceleration of the G1/S cell cycle, and by increasing the expression of iNOS, it contributes to tumor proliferation and metastasis. Following DNA damage and oxidative/heat stress, EGFR enters the cell nucleus and interacts with DNA-PK, leading to DNA repair mechanisms and radioresistance. RAS: rat sarcoma virus, BRAF: v-Raf murine sarcoma viral oncogene homolog B1, MEK: mitogen-activated protein kinase, ERK: extracellular signal-regulated kinase, MYC: myelocytoma proto-oncogene, PI3K: phosphatidylinositol 3-kinase, AKT: protein kinase B, mTOR: mammalian target of rapamycin, PLC-γ: phospholipase C, γ isoform, CaMK/PKC: calcium/calmodulin-dependent kinases/protein kinase C, STAT3: signal transducer and activator of transcription 3, PTEN: phosphatase and tensin homolog, E2F1: E2F transcription factor 1, DNA-PK: DNA-dependent protein kinase, PIP2: phosphatidylinositol 4,5-bisphosphate, IP3: inositol 1,4,5-trisphosphate, DAG: 1,2-diacylglycerol.

**Figure 2 cancers-17-00248-f002:**
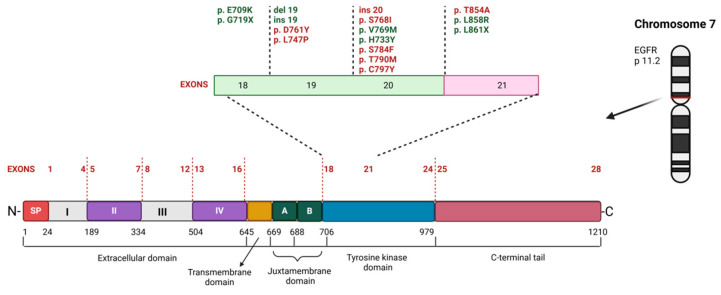
Schematic visualization of *EGFR* gene mutations in NSCLC (non-small-cell lung cancer) on the EGFR TKI (tyrosine kinase inhibitor) treatment. Location 18–21 of the most frequent *EGFR* mutations in NSCLC and their relationship to resistance (red) and sensitivity (green) in the treatment with EGFR TKIs. SP—signal peptide, del—deletion, ins—insertion.

**Figure 3 cancers-17-00248-f003:**
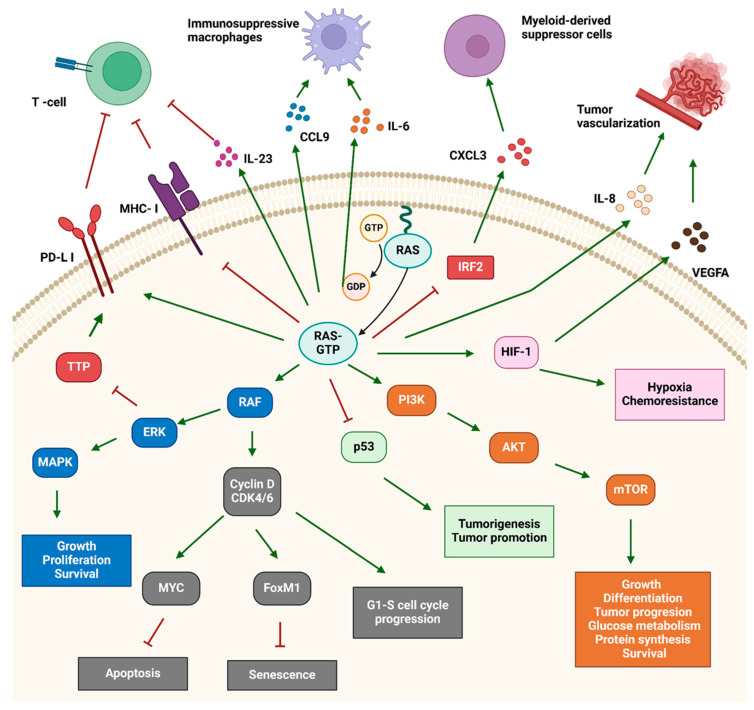
RAS triggering processes in cancer cells. Activated RAS performs various functions in the cell that promote cancer development, such as cell cycle progression, inhibition of apoptosis, cell growth and metabolism and cell motility and migration, and supports the evasion of cancer cells from the immune system. PD-L1: programmed cell death ligand 1, MHC-1: major histocompatibility complex 1, IL: interleucin, CCL9: chemokine (C-C motif) ligand 9, CXCL3: chemokine (C-X-C motif) ligand 3, VEGFA: vascular endothelial growth factor A, RAS: rat sarcoma virus, IRF2: interferon regulatory factor 2, HIF-1: hypoxia-inducible factor 1, TTP: tristetraprolin, RAF: rapidly accelerated fibrosarcoma kinases, ERK: extracellular signal-regulated kinase, MAPK: mitogen-activated protein kinases, PI3K: phosphatidylinositol 3-kinase, AKT: protein kinase B, mTOR: mammalian target of rapamycin, p53: tumor suppressor protein, CDK4/6: cyclin-dependent kinase 4 and 6, MYC: myelocytoma proto-oncogene, FoxM1: forkhead box M1, GTP: guanosinetriphosphate, GDP: guanosine diphosphate. RAS activates the MAPK pathways, which activate the transcription of several transcription factors (FOS, JUN, ETS, MYC…) that support the proliferation of cancer cells. RAS also plays an important role in activating the PI3K-AKT (phosphatidylinositol 3-kinase/protein kinase B) signaling pathway, which supports oncogenic transcription through the NF-κB (Nuclear factor-kappa B) signaling pathway, prevents apoptosis by inhibiting the pro-apoptotic protein BAD (BCL2 associated agonist of cell death), and promotes tumor cell growth and metabolism through mTOR (mammalian target of rapamycin) [34].

**Figure 4 cancers-17-00248-f004:**
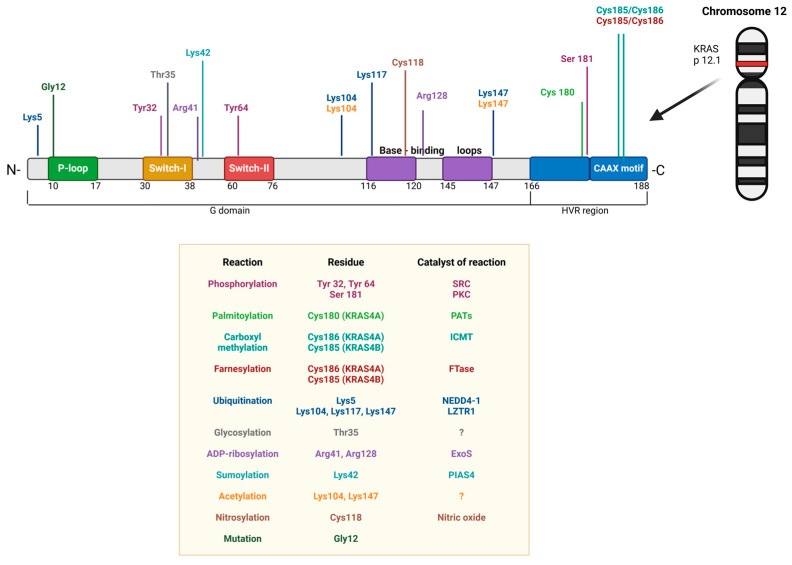
Schematic visualization of KRAS post-translational modification. Different modifications are distinguished by color in the yellow box (Phosphorylation—pink, Palmitoylation—green, Carboxylmetylation—dark green, Farnesylation—red, Ubiquitination—blue, Glycosylation—grey, ADP-ribosylation—purple, Sumolation—cyan, Acetylation—orange, Nitrosylation—brown). Switch 1 includes residues 25–40, and switch 2 includes residues 57–75, as well as the P-loop (phosphate-binding region) residues 10–17. The second lobe, comprised of residues 87–166, is a regulatory domain and contains allosteric regions such as helix 3 and loop. As found in all RAS GTPases, the HVR (hypervariable region) is located at the end of the second lobe, in K-Ras4B, comprising a 24-residue segment with the very C-terminal CAAX sequence (C, cysteine; A, aliphatic amino acid; X, any amino acid). ?—As yet unknown enzyme.

**Figure 5 cancers-17-00248-f005:**
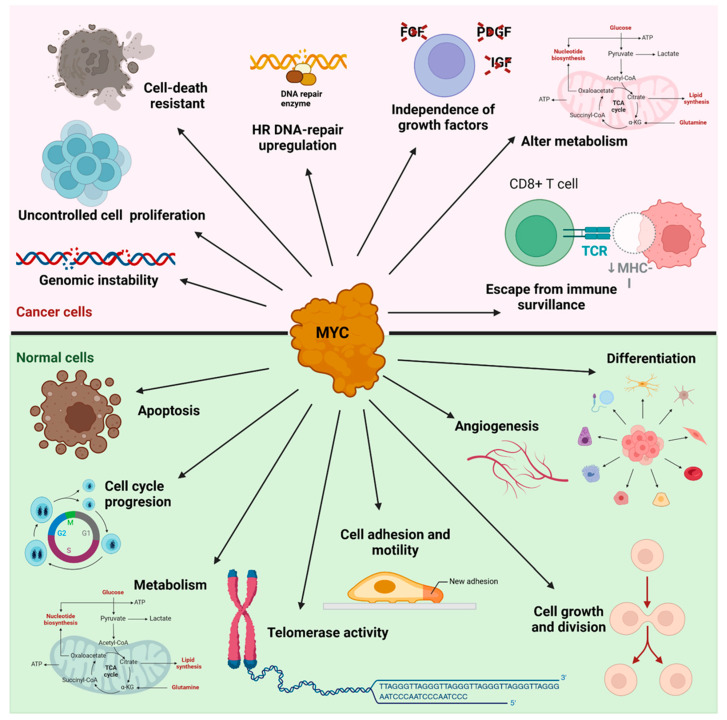
MYC-triggering processes in cancer and normal cells. Signaling processes activated in cancer (pink area) and normal cells (green area).

**Figure 6 cancers-17-00248-f006:**
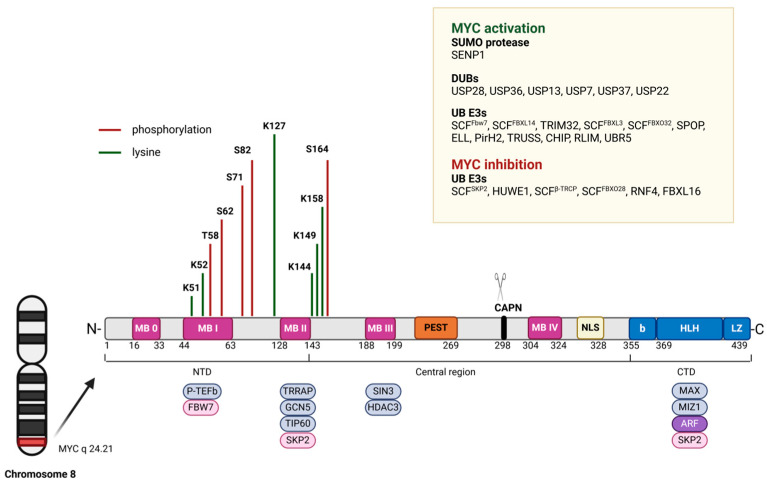
Schematic visualization of MYC post-translational modification. TAD (Trans Activation Domain), aa 1–150; I and II refer to conserved MYC boxes (MB), aa 44–63 and 128–143, respectively; A, acidic region (aa 242–261), NLS (Nuclear Localization Site) aa 320–328; bHLHZ, aa 355–435, which includes a basic region (B), helix-loop-helix (HLH), and leucine-zipper (LZ) domain that mediates binding to MAX and sequence-specific DNA binding; phosphorylation sites in MYC with relevant kinases (the red arrow indicates that Thr-58phosphorylation by GSK3 requires prior phosphorylation at Ser-62); Ac, acetylation sites with green arrows with relevant HAT enzymes (the following lysines are acetylated: 143, 148, 157, 275, 317, 323, 371, and 417). The central domain contains a PEST region. There are two conserved MBI and MBII motifs located in the TAD. Bottom: N (N-terminal domain), C (C-terminal domain), the principal PTMs, including phosphorylation(P-TEFb), acetylation (TRRAP—Transformation/transcription domain-associated protein, GCN5—General control non-depressible 5, TIP60—Tat-interactive protein, 60 kDa), ubiquitination (SKP2: S-phase kinase-associated protein 2, FBW7: F-Box and WD repeat domain containing 7), and sumoylation are reported in the yellow box legend; CAPN—calpain cleavage site. ARF2: ADP-ribosylation factor 2, MAX: Myc associated factor X, pink color: destabilization and blue color: stabilization properties.

**Table 1 cancers-17-00248-t001:** Some alterations in genes or proteins belonging to the EGFR/MYC/RAS axis affecting neoplastic cell proliferation.

Gene/Protein	Modification	Impact on Proliferation	Ref.
EGFR	deletion Ex19del	↑	[18]
	substitution L858R	↑	[19,22,23]
	hypermethylation 7 CpG a 17 CpG	↑	[26]
	methylation lysine 721	↑	[30]
RAS family	substitution G12C	↑	[44]
	substitution G12D	↑	[44]
	substitution G12R	↑	[44]
	substitution G12V	↑	[45]
	substitution Q61R	↑	[45]
	downregulation let-7	↑	[50,51,52]
	downregulation miR-181	↑	[54,55,57]
	targeting miR-18a-3p, miR-143 and miR-217	↓	[59,60,61]
	hypomethylation CpG islands in promotor	↑	[64]
	phosphorylation serine 181	?	[70]
	phosphorylation threonine 144 and 148	↓	[74,153]
	phosphorylation serine 89	↑	[75]
	monoubiquitination site 147	↑	[78,79]
	monoubiquitination site 117	↑	[81,82]
	sumolyation lysine 42	↑	[81,87]
	glucosylation threonine 35	↓	[91]
	nitrosylation cysteine 118	↓	[104]
MYC	gene amplification	↑	[119]
	hypomethylation CpG islands in promotor	↑	[122]
	acetylation lysine 323	↓	[129]
	deubiquitination—overexpression USP22	↓	[135]
	deubiquitination—overexpression USP36	↑	[140]
	deubiquitination—overexpression USP28	↑	[139]
	deubiquitination—overexpression USP37	↑	[136]
	phosphorylation serine 62	↑	[145]

Symbols: ↑—upregulation of cellular proliferation; ↓—downregulation of cellular proliferation; ?—ambiguous effect on cellular proliferation.

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
