# Peer review of "Dynamic Multilevel Regulation of EGFR, KRAS, and MYC Oncogenes: Driving Cancer Cell Proliferation Through (Epi)Genetic and Post-Transcriptional/Translational Pathways"

_cancers, 2025, doi:10.3390/cancers17020248_

Round 1

Reviewer 1 Report

Comments and Suggestions for Authors

1. The manuscript title suggests a focus on epigenetics; however, this is not adequately addressed and organised in the manuscript, which provides very limited information on epigenetics. Also, the study primarily focuses on EGFR-based cell proliferation. Therefore, I recommend that the authors revise the broad title to better reflect the specific scope and focus of the manuscript.

2. The manuscript needs to be reorganized into distinct sections for non-epigenetic and epigenetic content to ensure a consitant flow and improve readability.

3. Similar to Figure 4, the authors could consider providing a figure for the RAS pathway as well.

4. The authors should provide a table summarizing the factors and their roles in influencing EGFR/MYC/RAS-based cell proliferation.

5. There are several typographical errors in the manuscript regarding the formatting of gene and protein names. For example, gene names are not italicized in several places.

Author Response

Dear reviewer

Thank you for your comments that improve quality of our manuscript.

Yours

Authors

Reviewer 2 Report

Comments and Suggestions for Authors

Drs. Breier and Pavlíková have presented a timely and well-crafted review article examining the role of epigenetic and non-epigenetic regulation of oncogenes in cell proliferation. The article provides up-to-date information and incorporates recent discoveries in the field, offering significant translational potential for therapeutic applications.

However, a few key points should be addressed to improve the comprehensiveness and impact of the review:

  1. KRAS and Altered Metabolism: KRAS has been shown to regulate altered metabolism, which drives both epigenetic and non-epigenetic changes to promote uncontrolled cancer cell growth (PMID: 38311052). A discussion on this aspect should be included to enhance the article's depth.

  2. Hippo Signaling and KRAS Regulation: Under the section "Regulation of KRAS Oncogene Expression and Activity," the authors should explore the role of Hippo signaling. Recent studies reveal that Hippo signaling collaborates with KRAS signaling to overcome resistance to KRAS inhibitors (PMID: 37729426). Additionally, Hippo signaling is a key mediator of epigenetic regulation (PMID: 36864753), playing a significant role in driving both epigenetic and non-epigenetic changes in oncogenic KRAS signaling. Highlighting this pathway's therapeutic potential would add depth to the review.

  3. Clinical Trials: The authors should provide a list of relevant clinical trials associated with the topics discussed, offering a practical perspective on the translational implications of the research.

Addressing these points will enhance the comprehensiveness and impact of the review.

Author Response

Thank you for your recommendation that improve quality of our paper.

Reviewer 3 Report

Comments and Suggestions for Authors

The protooncogene families, RAS, myc and epidermal growth factor receptors, play a crucial role in oncogenesis and determine several hallmarks of cancer. The review by Mário Šereš and co-authors describes the important epigenetic changes in these genes that characterize tumorigenesis and shows which post-translational modifications affect the function of these proteins. This review provides a good snapshot of the regulation of oncogene activity and would be of interest to scientists in the field of cancer research. Although there are no major objections to the topic of the review, the manuscript is poorly prepared and suffers from multiple typographical errors (e.g., "Rasoncogene", "RASgene", etc.).  Authors should also check the punctuation throughout the text. The text in the "post translational modifications" panel in Figure 3 is not clear enough, probably due to the resolution problem. Could the authors improve the quality of the figures? In conclusion, the review is interesting but suffers from unfortunate errors.

Author Response

Thank you for your comments

We have tried to correct all errors and typos in the manuscript. We hope that we have made the manuscript more readable.

Yours authors

Reviewer 4 Report

Comments and Suggestions for Authors

This manuscript focuses on the problem of epigenetic changes and changes in post-translational modifications that have a major impact on the regulation/deregulation of oncogene expression levels. The authors analyze the impact of such changes on the EGFR-MYC-RAS signaling axis, because deregulated changes in this signaling pathway are common in many types of malignant diseases. The manuscript is well written and easy to read. Figures, especially 2 and 3, greatly facilitate the detailed illustration of the discussed findings. They also postulate that a more complete understanding of the nature of these changes will enable the application of therapy directed at changes in this axis. I found nothing in the text that would require changes or additions.

Author Response

Thank you for your comments

Your authors

Round 2

Reviewer 1 Report

Comments and Suggestions for Authors

Authors respond satisfactorily to my comments. 

Reviewer 2 Report

Comments and Suggestions for Authors

Accepted